# Deep Learning for Predicting Human Strategic Behavior

**Jason Hartford, James R. Wright, Kevin Leyton-Brown**
Department of Computer Science
University of British Columbia
{jasonhar, jrwright, kevinlb}@cs.ubc.ca

## Abstract

Predicting the behavior of human participants in strategic settings is an important problem in many domains. Most existing work either assumes that participants are perfectly rational, or attempts to directly model each participant's cognitive processes based on insights from cognitive psychology and experimental economics. In this work, we present an alternative, a deep learning approach that automatically performs cognitive modeling without relying on such expert knowledge. We introduce a novel architecture that allows a single network to generalize across different input and output dimensions by using matrix units rather than scalar units, and show that its performance significantly outperforms that of the previous state of the art, which relies on expert-constructed features.

## 1 Introduction

Game theory provides a powerful framework for the design and analysis of multiagent systems that involve strategic interactions [see, e.g., 16]. Prominent examples of such systems include search engines, which use advertising auctions to generate a significant portion of their revenues and rely on game theoretic reasoning to analyze and optimize these mechanisms [6, 20]; spectrum auctions, which rely on game theoretic analysis to carefully design the "rules of the game" in order to coordinate the reallocation of valuable radio spectrum [13]; and security systems, which analyze the allocation of security personnel as a game between rational adversaries in order to optimize their use of scarce resources [19]. In such applications, system designers optimize their choices with respect to assumptions about the preferences, beliefs and capabilities of human players [14]. A standard game theoretic approach is to assume that players are perfectly rational expected utility maximizers and indeed, that they have common knowledge of this. In some applications, such as the high-stakes spectrum auctions just mentioned, this assumption is probably reasonable, as participants are typically large companies that hire consultants to optimize their decision making. In other scenarios that allow less time for planning or involve less sophisticated participants, however, the perfect rationality assumption may lead to suboptimal system designs. For example, Yang et al. [24] were able to improve the performance of systems that defend against adversaries in security games by relaxing the perfect rationality assumption. Of course, relaxing this assumption means finding something else to replace it with: an accurate model of boundedly rational human behavior.

The behavioral game theory literature has developed a wide range of models for predicting human behavior in strategic settings by incorporating cognitive biases and limitations derived from observations of play and insights from cognitive psychology [2]. Like much previous work, we study the unrepeated, simultaneous-move setting, for two reasons. First, the setting is conceptually straightforward: games can be represented in a so-called "normal form", simply by listing the utilities to each player in for each combination of their actions (e.g., see Figure 1). Second, the setting is surprisingly general: auctions, security systems, and many other interactions can be modeled naturally

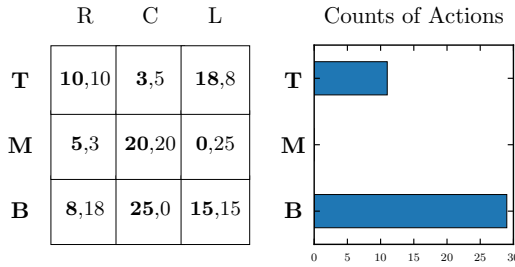

Figure 1: An example $3 \times 3$ normal form game. The row player chooses from actions $\{T, M, B\}$ and the column player chooses from actions $\{R, C, L\}$. If the row player played action $T$ and column player played action $C$, their resulting payoffs would be 3 and 5 respectively. Given such a matrix as input we aim to predict a distribution over the row player's choice of actions defined by the observed frequency of actions shown on the right.

as normal form games. The most successful predictive models for this setting combine notions of iterative reasoning and noisy best response [21] and use hand-crafted features to model the behavior of non-strategic players [23].

The recent success of deep learning has demonstrated that predictive accuracy can often be enhanced, and expert feature engineering dispensed with, by fitting highly flexible models that are capable of learning novel representations. A key feature in successful deep models is the use of careful design choices to encode "*basic* domain knowledge of the input, in particular its topological structure…to *learn* better features" [1, emphasis original]. For example, feed-forward neural nets can, in principle, represent the same functions as convolution networks, but the latter tend to be more effective in vision applications because they encode the prior that low-level features should be derived from the pixels within a small neighborhood and that predictions should be invariant to small input translations. Analogously, Clark and Storkey [4] encoded the fact that a Go board is invariant to rotations. These modeling choices constrain more general architectures to a subset of the solution space that is likely to contain good solutions. Our work seeks to do the same for the behavioral game theory setting, identifying novel prior assumptions that extend deep learning to predicting behavior in strategic scenarios encoded as two player, normal-form games.

A key property required of such a model is invariance to game size: a model must be able to take as input an $m \times n$ bimatrix game (i.e., two $m \times n$ matrices encoding the payoffs of players 1 and 2 respectively) and output an $m$-dimensional probability distribution over player 1's actions, for arbitrary values of $n$ and $m$, including values that did not appear in training data. In contrast, existing deep models typically assume either a fixed-dimensional input or an arbitrary-length sequence of fixed-dimensional inputs, in both cases with a fixed-dimensional output. We also have the prior belief that permuting rows and columns in the input (i.e., changing the order in which actions are presented to the players) does not change the output beyond a corresponding permutation. In Section 3, we present an architecture that operates on matrices using scalar weights to capture invariance to changes in the size of the input matrices and to permutations of its rows and columns. In Section 4 we evaluate our model's ability to predict distributions of play given normal form descriptions of games on a dataset of experimental data from a variety of experiments, and find that our feature-free deep learning model significantly exceeds the performance of the current state-of-the-art model, which has access to hand-tuned features based on expert knowledge [23].

## 2 Related Work

**Prediction in normal form games.** The task of predicting actions in normal form games has been studied mostly in the behavioral game theory literature. Such models tend to have few parameters and to aim to describe previously identified cognitive processes. Two key ideas are the relaxation of best response to "quantal response" and the notion of "limited iterative strategic reasoning". Models that assume *quantal response* assume that players select actions with probability increasing in expected utility instead of always selecting the action with the largest expected utility [12]. This is expressed formally by assuming that players select actions, $a_i$, with probability, $s_i$, given by the *logistic quantal response function* $s_i(a_i) = \frac{\exp(\lambda u_i(a_i, s_{-i}))}{\sum_{a_i'} \exp(\lambda u_i(a_i', s_{-i}))}$. This function is equivalent to the familiar softmax function with an additional scalar sharpness parameter $\lambda$ that allows the function to output the best response as $\lambda \to \infty$ and the uniform distribution as $\lambda \to 0$. This relaxation is motivated by the behavioral notion that if two actions have similar expected utility then they will also have similar probability of being chosen. *Iterative strategic reasoning* means that players perform a bounded

number of steps of reasoning in deciding on their actions, rather than always converging to fixed points as in classical game theory. Models incorporating this idea typically assume that every agent has an integer *level*. Non-strategic, "level-0" players choose actions uniformly at random; level-$k$ players best respond to the level-$(k-1)$ players [5] or to a mixture of levels between level-0 and level-$(k-1)$ [3]. The two ideas can be combined, allowing players to quantally respond to lower level players [18, 22]. Because iterative reasoning models are defined recursively starting from a base-case of level-0 behavior, their performance can be improved by better modeling the non-strategic level-0 players. Wright and Leyton-Brown [23] combine quantal response and bounded steps of reasoning with a model of non-strategic behavior based on hand-crafted game theoretic features. To the best of our knowledge, this is the current state-of-the-art model.

**Deep learning.** Deep learning has demonstrated much recent success in solving supervised learning problems in vision, speech and natural language processing [see, e.g., 9, 15]. By contrast, there have been relatively few applications of deep learning to multiagent settings. Notable exceptions are Clark and Storkey [4] and the policy network used in Silver et al. [17]'s work in predicting the actions of human players in Go. Their approach is similar in spirit to ours: they map from a description of the Go board at every move to the choices made by human players, while we perform the same mapping from a normal form game. The setting differs in that Go is a single, sequential, zero-sum game with a far larger, but fixed, action space, which requires an architecture tailored for pattern recognition on the Go board. In contrast, we focus on constructing an architecture that generalizes across general-sum, normal form games.

We enforce invariance to the size of the network's input. Fully convolutional networks [11] achieve invariance to the image size in a similar by manner replacing all fully connected layers with convolutions. In its architectural design, our model is mathematically similar to Lin et al. [10]'s Network in Network model, though we derived our architecture independently using game theoretic invariances. We discuss the relationships between the two models at the end of Section 3.

## 3 Modeling Human Strategic Behavior with Deep Networks

A natural starting point in applying deep networks to a new domain is testing the performance of a regular feed-forward neural network. To apply such a model to a normal form game, we need to flatten the utility values into a single vector of length $mn + nm$ and learn a function that maps to the $m$-simplex output via multiple hidden layers. Feed-forward networks can't handle size-invariant inputs, but we can temporarily set that problem aside by restricting ourselves to games with a fixed input size. We experimented with that approach and found that feed-forward networks often generalized poorly as the network overfitted the training data (see Section 2 of the supplementary material for experimental evidence). One way of combating overfitting is to encourage invariance through data augmentation: for example, one may augment a dataset of images by rotating, shifting and scaling the images slightly. In games, a natural simplifying assumption is that players are indifferent to the order in which actions are presented, implying invariance to permutations of the payoff matrix.[1] Incorporating this assumption by randomly permuting rows or columns of the payoff matrix at every epoch of training dramatically improved the generalization performance of a feed-forward network in our experiments, but the network is still limited to games of the size that it was trained on.

Our approach is to enforce this invariance in the model architecture rather than through data augmentation. We then add further flexibility using novel "pooling units" and by incorporating iterative response ideas inspired by behavioral game theory models. The result is a model that is flexible enough to represent the all the models surveyed in Wright and Leyton-Brown [22, 23]—and a huge space of novel models as well—and which can be identified automatically. The model is also invariant to the size of the input payoff matrix, differentiable end to end and trainable using standard gradient-based optimization.

The model has two parts: *feature layers* and *action response layers*; see Figure 2 for a graphical overview. The feature layers take the row and column player's normalized utility matrices $\mathbf{U}^{(r)}$ and $\mathbf{U}^{(c)} \in \mathbb{R}^{m \times n}$ as input, where the row player has $m$ actions and the column player has $n$ actions. The feature layers consist of multiple levels of *hidden matrix units*, $\mathbf{H}^{(r)}_{i,j} \in \mathbb{R}^{m \times n}$, each of which calculates a weighted sum of the units below and applies a non-linear activation function. Each

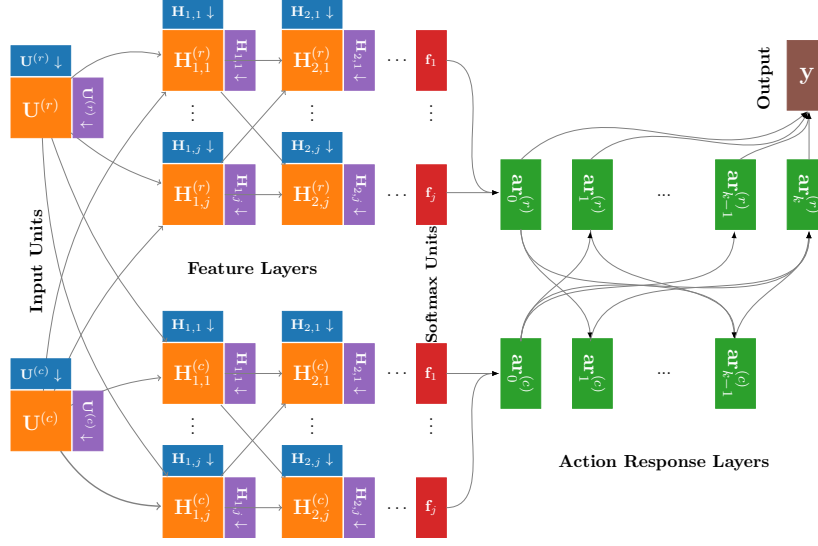

Figure 2: A schematic representation of our architecture. The feature layers consist of *hidden matrix units* (orange), each of which use *pooling units* to output row- and column-preserving aggregates (blue and purple) before being reduced to distributions over actions in the *softmax units* (red). Iterative response is modeled using the *action response layers* (green) and the final output, $y$, is a weighted sum of the row player's action response layers.

layer of hidden units is followed by *pooling units*, which output aggregated versions of the hidden matrices to be used by the following layer. After multiple layers, the matrices are aggregated to vectors and normalized to a distribution over actions, $\mathbf{f}_i^{(r)} \in \Delta^m$ in *softmax* units. We refer to these distributions as *features* because they encode higher-level representations of the input matrices that may be combined to construct the output distribution.

As discussed earlier, iterative strategic reasoning is an important phenomenon in human decision making; we thus want to allow our models the option of incorporating such reasoning. To do so, we compute features for the column player in the same manner by applying the feature layers to the transpose of the input matrices, which outputs $\mathbf{f}_i^{(c)} \in \Delta^n$. Each action response layer for a given player then takes the opposite player's preceding action response layers as input and uses them to construct distributions over the respective players' outputs. The final output $\mathbf{y} \in \Delta^m$ is a weighted sum of all action response layers' outputs.

**Invariance-Preserving Hidden Units** We build a model that ties parameters in our network by encoding the assumption that players reason about each action identically. This assumption implies that the row player applies the same function to each row of a given game's utility matrices. Thus, in a normal form game represented by the utility matrices $\mathbf{U}^{(r)}$ and $\mathbf{U}^{(c)}$, the weights associated with each row of $\mathbf{U}^{(r)}$ and $\mathbf{U}^{(c)}$ must be the same. Similarly, the corresponding assumption about the column player implies that the weights associated with each column of $\mathbf{U}^{(r)}$ and $\mathbf{U}^{(c)}$ must also be the same. We can satisfy both assumptions by applying a single scalar weight to each of the utility matrices, computing $w_r \mathbf{U}^{(r)} + w_c \mathbf{U}^{(c)}$. This idea can be generalized as in a standard feed-forward network to allow us to fit more complex functions. A hidden matrix unit taking all the preceding hidden matrix units as input can be calculated as

$$\mathbf{H}_{l,i} = \phi\left(\sum_j w_{l,i,j}\,\mathbf{H}_{l-1,j} + b_{l,i}\right) \quad \mathbf{H}_{l,i} \in \mathbb{R}^{m \times n},$$

where $\mathbf{H}_{l,i}$ is the $i^{th}$ hidden unit matrix for layer $l$, $w_{l,i,j}$ is the $j^{th}$ scalar weight, $b_{l,i}$ is a scalar bias variable, and $\phi$ is a non-linear activation function applied element-wise. Notice that, as in a traditional feed-forward neural network, the output of each hidden unit is simply a nonlinear transformation of the weighted sum of the preceding layer's hidden units. Our architecture differs by maintaining a

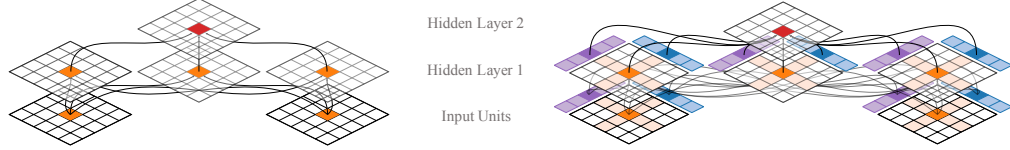

Figure 3: *Left:* Without pooling units, each element of every hidden matrix unit depends only on the corresponding elements in the units from the layer below; e.g., the middle element highlighted in red depends only on the value of the elements of the matrices highlighted in orange. *Right:* With pooling units at each layer in the network, each element of every hidden matrix unit depends both on the corresponding elements in the units below *and* the pooled quantity from each row and column. E.g., the light blue and purple blocks represent the row and column-wise aggregates corresponding to their adjacent matrices. The dark blue and purple blocks show which of these values the red element depends on. Thus, the red element depends on both the dark- and light-shaded orange cells.

matrix at each hidden unit instead of a scalar. So while in a traditional feed-forward network each hidden unit maps the previous layer's vector of outputs into a scalar output, in our architecture each hidden unit maps a tensor of outputs from the previous layer into a matrix output.

Tying weights in this way reduces the number of parameters in our network by a factor of $nm$, offering two benefits. First, it reduces the degree to which the network is able to overfit; second and more importantly, it makes the model invariant to the size of the input matrices. To see this, notice that each hidden unit maps from a tensor containing the $k$ output matrices of the preceding layer in $\mathbb{R}^{k \times m \times n}$ to a matrix in $\mathbb{R}^{m \times n}$ using $k$ weights. Thus our number of parameters in each layer depends on the number of hidden units in the preceding layer, but not on the sizes of the input and output matrices. This allows the model to generalize to input sizes that do not appear in training data.

**Pooling units** A limitation of the weight tying used in our hidden matrix units is that it forces independence between the elements of their matrices, preventing the network from learning functions that compare the values of related elements (see Figure 3 (left)). Recall that each element of the matrices in our model corresponds to an outcome in a normal form game. A natural game theoretic notion of the "related elements" which we'd like our model to be able to compare is the set of payoffs associated with each of the players' actions that led to that outcome. This corresponds to the row and column of each matrix associated with the particular element.

This observation motivates our pooling units, which allow information sharing by outputting aggregated versions of their input matrix that may be used by later layers in the network to learn to compare the values of a particular cell in a matrix and its row- or column-wise aggregates.

$$\mathbf{H} \to \{\mathbf{H}_c, \mathbf{H}_r\} = \left\{ \begin{pmatrix} \max_i h_{i,1} & \max_i h_{i,2} & \dots \\ \max_i h_{i,1} & \max_i h_{i,2} & \dots \\ \vdots & \vdots & \\ \max_i h_{i,1} & \max_i h_{i,2} & \end{pmatrix}, \begin{pmatrix} \max_j h_{1,j} & \max_j h_{1,j} & \dots \\ \max_j h_{2,j} & \max_j h_{2,j} & \dots \\ \vdots & \vdots & \\ \max_j h_{m,j} & \max_j h_{m,j} & \dots \end{pmatrix} \right\} \quad (1)$$

A pooling unit takes a matrix as input and outputs two matrices constructed from row- and column-preserving pooling operations respectively. A pooling operation could be any continuous function that maps from $\mathbb{R}^n \to \mathbb{R}$. We use the *max* function because it is a necessary to represent known behavioral functions (see Section 4 of the supplementary material for details) and offered the best empirical performance of the functions we tested. Equation (1) shows an example of a pooling layer with *max* functions for some arbitrary matrix $\mathbf{H}$. The first of the two outputs, $\mathbf{H}_c$, is column-preserving in that it selects the maximum value in each column of $\mathbf{H}$ and then stacks the resulting vector $n$-dimensional vector $m$ times such that the dimensionality of $\mathbf{H}$ and $\mathbf{H}_c$ are the same. Similarly, the row-preserving output constructs a vector of the max elements in each column and stacks the resulting $m$-dimensional vector $n$ times such that $\mathbf{H}_r$ and $\mathbf{H}$ have the same dimensionality. We stack the vectors that result from the pooling operation in this fashion so that the hidden units from the next layer in the network may take $\mathbf{H}, \mathbf{H}_c$ and $\mathbf{H}_r$ as input. This allows these later hidden units to learn functions where each element of their output is a function both of the corresponding element from the matrices below as well as their row and column-preserving maximums (see Figure 3 (right)).

**Softmax output**    Our model predicts a distribution over the row player's actions. In order to do this, we need to map from the hidden matrices in the final layer, $\mathbf{H}_{L,i} \in R^{m \times n}$, of the network onto a point on the $m$-simplex, $\Delta^m$. We achieve this mapping by applying a row-preserving sum to each of the final layer hidden matrices $\mathbf{H}_{L,i}$ (i.e. we sum uniformly over the columns of the matrix as described above) and then applying a softmax function to convert each of the resulting vectors $\mathbf{h}_i$ into normalized distributions. This produces $k$ features $\mathbf{f}_i$, each of which is a distribution over the row player's $m$ actions:

$$\mathbf{f}_i = \text{softmax}\left(\mathbf{h}^{(i)}\right) \quad \text{where } \mathbf{h}_j^{(i)} = \sum_{k=1}^{n} h_{j,k}^{(i)} \text{ for all } j \in \{1,...,m\}, \, h_{j,k}^{(i)} \in \mathbf{H}^{(i)} \quad i \in \{1,...,k\}.$$

We can then produce the output of our features, $\mathbf{ar}_0$, using a weighted sum of the individual features, $\mathbf{ar}_0 = \sum_{i=1}^{k} w_i \mathbf{f}_i$, where we optimize $w_i$ under simplex constraints, $w_i \geq 0, \sum_i w_i = 1$. Because each $\mathbf{f}_i$ is a distribution and our weights $w_i$ are points on the simplex, the output of the feature layers is a mixture of distributions.

**Action Response Layers**    The feature layers described above are sufficient to meet our objective of mapping from the input payoff matrices to a distribution over the row player's actions. However, this architecture is not capable of explicitly representing iterative strategic reasoning, which the behavioral game theory literature has identified as an important modeling ingredient. We incorporate this ingredient using action response layers: the first player can respond to the second's beliefs, the second can respond to this response by the first player, and so on to some finite depth. The proportion of players in the population who iterate at each depth is a parameter of the model; thus, our architecture is also able to learn not to perform iterative reasoning.

More formally, we begin by denoting the output of the feature layers as $\mathbf{ar}_0^{(r)} = \sum_{i=1}^{k} w_{0i}^{(r)} \mathbf{f}_i^{(r)}$, where we now include an index $(r)$ to refer to the output of *row* player's action response layer $\mathbf{ar}_0^{(r)} \in \Delta^m$. Similarly, by applying the feature layers to a transposed version of the input matrices, the model also outputs a corresponding $\mathbf{ar}_0^{(c)} \in \Delta^n$ for the column player which expresses the row player's beliefs about which actions the column player will choose. Each action response layer composes its output by calculating the expected value of an internal representation of utility with respect to its belief distribution over the opposition actions. For this internal representation of utility we chose a weighted sum of the final layer of the hidden layers, $\sum_i w_i \mathbf{H}_{L,i}$, because each $\mathbf{H}_{L,i}$ is already some non-linear transformation of the original payoff matrix, and so this allows the model to express utility as a transformation of the original payoffs. Given the matrix that results from this sum, we can compute expected utility with respect to the vector of beliefs about the opposition's choice of actions, $\mathbf{ar}_j^{(c)}$, by simply taking the dot product of the weighted sum and beliefs. When we iterate this process of responding to beliefs about one's opposition more than once, higher-level players will respond to beliefs, $\mathbf{ar}_i$, for all $i$ less than their level and then output a weighted combination of these responses using some weights, $v_{l,i}$. Putting this together, the $l^{th}$ action response layer for the *row* player $(r)$ is defined as

$$\mathbf{ar}_l^{(r)} = \text{softmax}\left( \lambda_l \left( \sum_{j=0}^{l-1} v_{l,j}^{(r)} \left( \sum_{i=1}^{k} w_{l,i}^{(r)} \mathbf{H}_{L,i}^{(r)} \right) \cdot \mathbf{ar}_j^{(c)} \right) \right), \quad \mathbf{ar}_l^{(r)} \in \Delta^m, l \in \{1,...,K\},$$

where $l$ indexes the action response layer, $\lambda_l$ is a scalar sharpness parameter that allows us to sharpen the resulting distribution, $w_{l,i}^{(r)}$ and $v_{l,j}^{(r)}$ are scalar weights, $\mathbf{H}_{L,i}$ are the *row* player's $k$ hidden units from the final hidden layer $L$, $\mathbf{ar}_j^{(c)}$ is the output of the *column* player's $j^{th}$ action response layer, and $K$ is the total number of action response layers. We constrain $w_{li}^{(r)}$ and $v_{lj}^{(r)}$ to the simplex and use $\lambda_l$ to sharpen the output distribution so that we can optimize the sharpness of the distribution and relative weighting of its terms independently. We build up the column player's action response layer, $\mathbf{ar}_l^{(c)}$, similarly, using the column player's internal utility representation, $\mathbf{H}_{L,i}^{(c)}$, responding to the row player's action response layers, $\mathbf{ar}_l^{(r)}$. These layers are not used in the final output directly but are relied upon by subsequent action response layers of the row player.

**Output**    Our model's final output is a weighted sum of the outputs of the action response layers. This output needs to be a valid distribution over actions. Because each of the action response layers

also outputs a distribution over actions, we can achieve this requirement by constraining these weights to the simplex, thereby ensuring that the output is just a mixture of distributions. The model's output is thus $\mathbf{y} = \sum_{j=1}^{K} w_j \mathbf{ar}_j^{(r)}$, where $\mathbf{y}$ and $\mathbf{ar}_j^{(r)} \in \Delta^m$, and $w_j \in \Delta^K$.

**Relation to existing deep models**   Our model's functional form has interesting connections with existing deep model architectures. We discuss two of these here. First, our invariance-preserving hidden layers can be encoded as *MLP Convolution Layers* described in Lin et al. [10] with the two-channel $1 \times 1$ input $x_{i,j}$ corresponding to the two players' respective payoffs when actions $i$ and $j$ are played (using patches larger than $1 \times 1$ would imply the assumption that local structure is important, which is inappropriate in our domain; thus, we do not need multiple *mlpconv* layers). Second, our pooling units are superficially similar to the pooling units used in convolutional networks. However, ours differ both in functional form and purpose: we use pooling as a way of sharing information between cells in the matrices that are processed through our network by taking maximums across entire rows or columns, while in computer vision, max-pooling units are used to produce invariance to small translations of the input image by taking maximums in a small local neighborhood.

**Representational generality of our architecture**   Our work aims to extend existing models in behavioral game theory via deep learning, not to propose an orthogonal approach. Thus, we must demonstrate that our representation is rich enough to capture models and features that have proven important in that literature. We omit the details here for space reasons (see the supplementary material, Section 4), but summarize our findings. Overall, our architecture can express the *quantal cognitive hierarchy* [23] and *quantal level-k* [18] models and as their sharpness tends to infinity, their best-response equivalents *cognitive hierarchy* [3] and *level-k* [5]. Using feature layers we can also encode all the behavioral features used in Wright and Leyton-Brown [23]. However, our architecture is not universal; notably, it is unable to express certain features that are likely to be useful, such as identification of dominated strategies. We plan to explore this in future work.

# 4   Experiments

**Experimental Setup**   We used a dataset combining observations from 9 human-subject experimental studies conducted by behavioral economists in which subjects were paid to select actions in normal-form games. Their payment depended on the subject's actions and the actions of their unseen opposition who chose an action simultaneously (see Section 1 of the supplementary material for further details on the experiments and data). We are interested in the model's ability to predict the distribution over the row player's action, rather than just its accuracy in predicting the most likely action. As a result, we fit models to maximize the likelihood of training data $\mathbb{P}(\mathcal{D}|\theta)$ (where $\theta$ are the parameters of the model and $\mathcal{D}$ is our dataset) and evaluate them in terms of negative log-likelihood on the test set.

All the models presented in the experimental section were optimized using Adam [8] with an initial learning rate of 0.0002, $\beta_1 = 0.9$, $\beta_2 = 0.999$ and $\epsilon = 10^{-8}$. The models were all regularized using Dropout with drop probability $= 0.2$ and $L_1$ regularization with parameter $= 0.01$. They were all trained until there was no training set improvement up to a maximum of 25 000 epochs and the parameters from the iteration with the best training set performance was returned. Our architecture imposes simplex constraints on the mixture weight parameters. Fortunately, simplex constraints fall within the class of *simple constraints* that can be efficiently optimized using the projected gradient algorithm [7]. The algorithm modifies standard SGD by projecting the relevant parameters onto the constraint set after each gradient update.

**Experimental Results**   Figure 4 (left) shows a performance comparison between a model built using our deep learning architecture with only a single action response layer (i.e. no iterative reasoning; details below) and the previous state of the art, quantal cognitive hierarchy (QCH) with hand-crafted features (shown as a blue line); for reference we also include the best feature-free model, QCH with a uniform model of level-0 behavior (shown as a pink line). We refer to an instantiation of our model with $L$ hidden layers and $K$ action response layers as an $N + K$ layer network. All instantiations of our model with 3 or more layers significantly improved on both alternatives and thus represents a new state of the art. Notably, the magnitude of the improvement was considerably larger than that of adding hand-crafted features to the original QCH model.

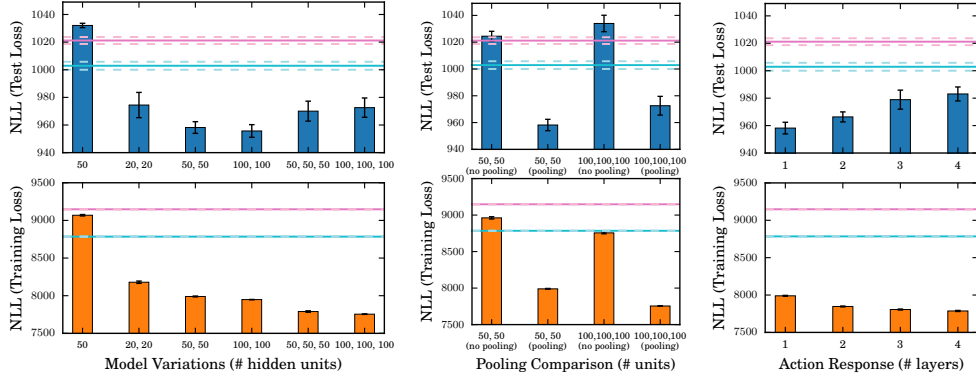

Figure 4: Negative Log Likelihood Performance. The error bars represent 95% confidence intervals across 10 rounds of 10-fold cross-validation. We compare various models built using our architecture to QCH Uniform (pink line) and QCH Linear4 (blue line).

Figure 4 (left) considers the effect of varying the number of hidden units and layers on performance using a single action response layer. Perhaps unsurprisingly, we found that a two layer network with only a single hidden layer of 50 units performed poorly on both training and test data. Adding a second hidden layer resulted in test set performance that improved on the previous state of the art. For these three layer networks (denoted (20, 20), (50, 50) and (100, 100)), performance improved with more units per layer, but there were diminishing returns to increasing the number of units per layer beyond 50. The four-layer networks (denoted (50, 50, 50) and (100, 100, 100)) offered further improvements in training set performance but test set performance diminished as the networks were able to overfit the data. To test the effect of pooling units on performance, in Figure 4 (center) we first removed the pooling units from two of the network configurations, keeping the rest of the hyper-parameters unchanged. The models that did not use pooling layers under fit on the training data and performed very poorly on the test set. While we were able to improve their performance by turning off dropout, these unregularized networks didn't match the training set performance of the corresponding network configurations that had pooling units (see Section 3 of the supplementary material). Thus, our final network contained two layers of 50 hidden units and pooling units.

Our next set of experiments committed to this configuration for feature layers and investigated configurations of action-response layers, varying their number between one and four (i.e., from no iterative reasoning up to three levels of iterative reasoning; see Figure 4 (right) ). The networks with more than one action-response layer showed signs of overfitting: performance on the training set improved steadily as we added AR layers but test set performance suffered. Thus, our final network used only one action-response layer. We nevertheless remain committed to an architecture that can capture iterative strategic reasoning; we intend to investigate more effective methods of regularizing the parameters of action-response layers in future work.

## 5   Discussion and Conclusions

To design systems that efficiently interact with human players, we need an accurate model of boundedly rational behavior. We present an architecture for learning such models that significantly improves upon state-of-the-art performance without needing hand-tuned features developed by domain experts. Interestingly, while the full architecture can include action response layers to explicitly incorporate the iterative reasoning process modeled by level-$k$-style models, our best performing model did not need them to achieve set a new performance benchmark. This indicates that the model is performing the mapping from payoffs to distributions over actions in a manner that is substantially different from previous successful models. Some natural future directions, besides those already discussed above, are to extend our architecture beyond two-player, unrepeated games to games with more than two players, as well as to richer interaction environments, such as games in which the same players interact repeatedly and games of imperfect information.

## Footnotes

[1]We thus ignore salience effects that could arise from action ordering; we plan to explore this in future work.

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
