[Supplementary Material]

# Deep Learning for Predicting Human Strategic Behavior - Supplementary Material

**Jason Hartford, James R. Wright, Kevin Leyton-Brown**
Department of Computer Science
University of British Columbia
{jasonhar, jrwright, kevinlb}@cs.ubc.ca

## 1  Data

Table 1: Our datasets. Each experiment had subjects play between 8 and 20 games for a total of 128 games, 113 of which were unique.

| Source | Games | $n$ |
|---|---|---|
| Stahl and Wilson [1994] | 10 | 400 |
| Stahl and Wilson [1995] | 12 | 576 |
| Costa-Gomes et al. [1998] | 18 | 1296 |
| Goeree and Holt [2001] | 10 | 500 |
| Cooper and Huyck [2003] | 8 | 2992 |
| Rogers et al. [2009] | 17 | 1210 |
| Haruvy et al. [2001] | 15 | 869 |
| Haruvy and Stahl [2007] | 20 | 2940 |
| Stahl and Haruvy [2008] | 18 | 1288 |
| All9 | 113 unique | 12071 |

We used a dataset that combined observations from 9 human-subject experimental studies conducted by behavioural economists in which subjects were paid to select actions in normal-form games. Their payment depended on the subject's actions and the actions of their unseen opposition who chose an action simultaneously. The subjects were shown the payment for each outcome using a payoff matrix that lists each pair of actions and the respective player's payments. Each experiment presented subjects with between 8 and 20 different games and the number of subjects who selected each action is recorded. Obtained outcomes were only shown at the end of each experiment to prevent learning effects. The games range in size from $2 \times 2$ to $121 \times 121$; in the majority, players have three actions each.

Our model takes such payoff matrices as input and predicts the observed frequency with which each action was selected by the row player. We encode the payoffs of each normal form game as a pair of utility matrices corresponding to the respective players' payoffs, normalized such that the standard deviation of the payoffs is approximately 1. For symmetric games we combine observations as though both players were the row player and for asymmetric games we treat observations of the column player's choice of actions as though they had come from a game with a transposed payoff matrix, such that they become the row player. A few games appear in multiple experiments; we combined their observed frequencies into a single common game.

We evaluate performance of the models using 10-fold cross-validation. We randomly partitioned our 113 unique games into 10 folds containing between 11 and 15 games each. Our experimental results examine both the mean and variance across 10 different such 10-fold cross-validations, each time re-randomizing the assignment of games into folds (requiring each model to be retrained 100 times).

We are interested in the model's ability to predict the distribution over the row player's action, rather than just its accuracy in predicting the most likely action. As a result, we fit models to maximize the likelihood of training data $\mathbb{P}(\mathcal{D}|\theta)$ (where $\theta$ are the parameters of the model and $\mathcal{D}$ is our dataset) and evaluate them in terms of negative log-likelihood on the test set.

## 2 Regular neural network performance

Figure 1: Performance comparison on $3 \times 3$ games of a feed forward neural network (FFNet), a feed forward neural network with data augmentation at every epoch (FFNet (Permuted)), our architecture fit with the same hyper parameters as used for our best performing model in the main results (GameNet), and Quantal Cognitive Hierarchy with four hand-crafted features (QCH Linear4).

Figure 1 compares the performance of our architecture with that of a regular feed-forward neural network, with and without data augmentation, and the previous state-of-the-art model on this dataset. It shows that the feed-forward network dramatically overfitted the data without data augmentation. Data augmentation improved test set performance, but it was still unable to match state of the art performance. A three layer instantiation of our model (two layers of 50 hidden units and a single AR layer) matched the previous state of the art but failed to improve upon it. We suspect that this may be because the subset of the data that contains only $3 \times 3$ games is too small to take advantage of the flexibility of our model.

## 3 Pooling units performance

To test the effect of pooling units on performance, in Figure 2 we first removed the pooling units from two of the network configurations, keeping the rest of the hyperparameters unchanged. The models that did not use pooling layers underfit on the training data and performed very poorly on the

Figure 2: Performance comparison with and without pooling units. All models were fit with the same hyperparameters using dropout unless otherwise stated, with the only difference being the number of layers and hidden units and whether or not the models used pooling units.

test set. While we were able to improve their performance by turning off dropout, these unregularised networks didn't match the training set performance of the corresponding network configurations that had pooling units. The test set performance for all the networks we trained without pooling units remained significantly worse than our best performing networks that used pooling units. Thus, our final network contained two layers of 50 hidden units and pooling units and used dropout.

# 4    Representational ability of our network

In this section we make explicit the connection between our model and popular models from the behavioral game theory literature by demonstrating how our architecture is able to express these models. At a high level, we express behavioral models using appropriately parameterized action response layers and we express non-strategic features using the invariance preserving hidden layers with pooling units.

## 4.1    Models

The four behavioral models we consider are *quantal cognitive hierarchy* (QCH) [Wright and Leyton-Brown, 2014, Stahl and Wilson, 1994] and *quantal level-$k$* (QLk) [Stahl and Wilson, 1994], *cognitive hierarchy* (CH) [Camerer et al., 2004] and *level-$k$* (Lk) [Costa-Gomes et al., 2001]. They differ in their behavioral assumptions, but they are similar in their mathematical descriptions. All involve some notion of a *response* (in the form of a *strategy* or distribution over one's own actions) to *beliefs* about one's opposition strategy (in the form of a distribution over one's opposition's actions).

**Responses**    We can divide these four models into two distinct classes: either they assume players *best respond* or *quantally respond*. Players best respond by selecting an action from the set of actions that maximizes their expected utility given their beliefs. Alternatively, they quantally respond by choosing actions with probability proportional to the action's expected utility.

This is modeled formally as follows: let $\bar{u}_i(\mathbf{s}) = \sum_{j=1}^n u_{i,j} s_j$ denote a player's expected utility, given beliefs, $\mathbf{s}$, about the opposition actions. A quantal response strategy is defined as

$$s_i(\mathbf{s_j}) = \frac{\exp(\lambda \bar{u}_i(\mathbf{s}))}{\sum_{i=1}^m \exp(\lambda \bar{u}_i(\mathbf{s}))}.$$

Quantal response approaches best response as $\lambda \to \infty$ in the sense that it defines a strategy where players uniformly randomize from the set of best responses[1]. Alternatively, if $\lambda \to 0$, players ignore their payoffs and uniformly randomize over their set of actions.

**Beliefs**  The four models can also be categorized based on how they define beliefs about one's opposition, $\mathbf{s}_j$. All of the models rely on a notion of a "cognitive level" that differs among players. However, while Level-$k$ models assume that a player at cognitive level $k$ only has beliefs about level $(k-1)$ players, cognitive hierarchy models assume that players respond to beliefs about the full distribution of players having cognitive level less than their own.

We now show the connection between our neural network-based approach and these behavioral models. Recall that action response layer $l$ is defined as

$$\mathbf{ar}_l^{(r)} = \text{softmax}\left(\lambda_l \left(\sum_{j=0}^{l-1} v_{l,j}^{(r)} \left(\sum_{i=1}^k w_{l,i}^{(r)} \mathbf{H}_{L,i}^{(r)}\right) \cdot \mathbf{ar}_j^{(c)}\right)\right), \quad \mathbf{ar}_l^{(r)} \in \Delta^m, l \in \{1, ..., K\},$$

and action response layer 0 is defined as a weighted sum of features, $\mathbf{ar}_0 = \sum_{i=1}^k w_i \mathbf{f}_i$.

The behavioral models do not depend on transformed versions of the input matrices or behavioral features, so we let the parameters of the network be set such that

$$\mathbf{ar}_l^{(r)} = \text{softmax}\left(\lambda_l \left(\sum_{j=0}^{l-1} v_{l,j}^{(r)} \mathbf{U}^{(r)} \cdot \mathbf{ar}_j^{(c)}\right)\right), \quad \mathbf{ar}_l^{(r)} \in \Delta^m, l \in \{1, ..., K\},$$

and let $\mathbf{ar}_{0,i} = \frac{1}{m}$ for all $i \in \{1, \ldots, m\}$.

## 4.2  Level-$k$

The Level-$k$ model associates every player in the population with a particular cognitive level corresponding to the number of steps of strategic reasoning they complete (bounded by some fixed maximum level), and assumes that level-$k$ players best respond to the strategy played by level-$(k-1)$ players and that level-0 players select actions uniformly at random. Each level also has some probability $\epsilon_k$ of making an "error" by selecting some action other than their best response.

We model this with the action response layers, by setting

$$v_{l,i}^{(r)} = \begin{cases} 1 - \epsilon_i & \text{if } i = l - 1 \\ \epsilon_i & \text{if } i = 0 \\ 0 & \text{otherwise.} \end{cases}$$

and letting $\lambda_l \to \infty$ in order to simulate best response.

## 4.3  Cognitive Hierarchy

Cognitive Hierarchy is similar to level-$k$ except it assumes a distribution over the levels a player may take, and assumes they best respond without error to the normalized distribution of players below them.

That is, there is some level distribution $\mathbf{p}$ where $p_i$ is the proportion of players in the population who behave according to a particular level and players respond to a normalized distribution $\mathbf{p}_{[0:i-1]}$.

We model this with the action response layers, by setting

$$v_{l,i}^{(r)} = \frac{p_i}{\sum_{j=0}^{i-1} p_j}$$

and letting $\lambda_l \to \infty$ in order to simulate best response.

## 4.4 Quantal Level-$k$

Quantal Level-$k$ differs from the level-$k$ model described above by allowing players to quantally respond with each level having a different precision parameter $\lambda_l$. The parameters remain as described in Section 4.2 except that we use $\lambda_l$ instead of letting $\lambda \to \infty$

## 4.5 Quantal Cognitive Hierarchy

Quantal Cognitive Hierarchy [Stahl and Wilson, 1994, Wright and Leyton-Brown, 2014] generalists cognitive hierarchy by allowing players to quantally respond and optimism the parameter $\lambda$. Similarly to the above, the parameters of our action response layers remain the same as in Section 4.3 except we use $\lambda_l = \lambda$ instead of letting $\lambda_l \to \infty$.

## 4.6 Game Theoretic Features

Wright and Leyton-Brown [2014] showed the importance of explicitly modeling nonstrategic level-0 players. They investigated a wide range of models and found that weighted linear combinations of nonstrategic features were most effective for improving predictive performance. In this section we argue that our model is sufficiently flexible to represent all the behavioral features used in their best performing models which allows us to generalize the quantal cognitive hierarchy with weighted linear features model presented in Wright and Leyton-Brown [2014]. We make this claim formally below.

**Claim:** A network with two hidden layers, one hidden unit per layer, pooling units at every layer and rectified linear unit activation functions can represent each of the following normalised features,

- *min max regret*,
- *min min unfairness*,
- *max min payoff*,
- *max max payoff*
- *max max efficiency*.

**Proof** By expanding the sums from the definition of the network, we see the first hidden layer has the following functional form:

$$\mathbf{H}^{(1,1)} = \mathrm{relu}(w_{1,r}\mathbf{U}^{(r)} + w_{1,c}\mathbf{U}^{(c)} + w_{1,rc}\mathbf{U}_c^{(r)} + w_{1,rr}\mathbf{U}_r^{(r)} + w_{1,cc}\mathbf{U}_c^{(c)} + w_{1,cr}\mathbf{U}_r^{(c)} + b_{1,1}).$$

where $\mathbf{U}^{(r)}$ is the row player's payoff matrix and $\mathbf{U}_c^{(r)}$ is the row player's payoff matrix aggregated using the column-preserving pooling unit where we use the max function to perform the aggregation. Similarly, the second hidden layer can be written as,

$$\mathbf{H}^{(2,1)} = \mathrm{relu}(w_{2,1}\mathbf{H}^{(1,1)} + w_{2,c}\mathbf{H}_c^{1,1} + w_{2,r}\mathbf{H}_r^{(1,1)} + b_{2,1}).$$

We denote $\mathbf{H}^{(1,1)}$ as the output of the first hidden layer and $\mathbf{H}_c^{(1,1)}$ and $\mathbf{H}_r^{(1,1)}$ are its respective pooled outputs.

Game theoretic features can be interpreted as outputting a strategy (a distribution over a player's actions) given a description over the game. We express features in a style similar to [Wright and Leyton-Brown, 2014] by outputting a vector $\mathbf{f}$ such that $f_i \approx 0$ for all $f_i \in \mathbf{f}$ if action $i$ does not correspond to the target feature, and $f_i \approx \frac{1}{l}$ where $l$ is the number of actions that correspond to the target feature (with $l = 1$ if the actions uniquely satisfies the feature; Wright and Leyton-Brown [2014] instead used a binary encoding, but that does not fit naturally into our framework). We have approximate equality, $\approx$, because we construct the features using a softmax function and hence our output approaches $f_i = 0$ or $\frac{1}{l}$ as our parameters $\to \infty$. Because features are all constructed from a sparse subset of the parameters, we limit notational complexity by letting $w_{i,j} = 0$ and $b_{i,j} = 0$ for all $i, j \in 1, 2, r, c$ unless stated otherwise.

### 4.7 Max Max Payoff

**Required:** $f^{\text{maxmax}}(i) = \begin{cases} \frac{1}{l} & \text{if } i \in \arg\max_{i \in \{1,\ldots,m\}} \max_{j \in \{1,\ldots,n\}} u_{i,j} \quad u_{i,j} \in \mathbf{U}^{(r)} \\ 0 & \text{otherwise} \end{cases}$

Let $w_{1,r} = 1, w_{2,r} = c$ where c is some large positive constant and $b_{1,1} = b$ where is some scalar $b \geq \min_{i,j} \mathbf{U}^{(r)}_{i,j}$ and all other parameters $w_{i,j}, b_{i,j} = 0$. Then $\mathbf{H}^{(1,1)}$ reduces to,

$$\mathbf{H}^{(1,1)} = \text{relu}(\mathbf{U}^{(r)} + b) = \mathbf{U}^{(r)} + b \quad \text{since } \mathbf{U}^{(r)} + b \geq 0 \text{ by definition of b}$$

$$\mathbf{H}^{(2,1)} = \text{relu}(c\mathbf{H}^{(1,1)}_r) \Rightarrow h_{j,k} = c(\max_k u_{j,k} + b) \quad \forall u_{j,k} \in \mathbf{U}^{(r)}, h_{j,k} \in \mathbf{H}^{(2,1)}$$

That is, all the elements in each row of $\mathbf{H}^{(2,1)}$ equal an positive affine transformation of the maximum element from the corresponding row in $\mathbf{U}^{(r)}$.

$$\mathbf{f}^{(1)}_i = \text{softmax}(\sum_{k=1}^{n} h_{j,k}) = \text{softmax}\left(\sum_{k=1}^{n} c(\max_k u_{j,k} + b)\right) = \text{softmax}\left(nc(\max_k u_{j,k} + b)\right)$$

Therefore, as $c \to \infty$, $\mathbf{f}^{(1)}_i \to f^{maxmax}(i)$ as required.

### 4.8 Max Min Payoff

**Required:** $f^{\text{maxmin}}(i) = \begin{cases} \frac{1}{l} & \text{if } i \in \arg\max_{i \in \{1,\ldots,m\}} \min_{j \in \{1,\ldots,n\}} u_{i,j} \quad u_{i,j} \in \mathbf{U}^{(r)} \\ 0 & \text{otherwise} \end{cases}$

Max Min Payoff is derived similarly to Max Max except with $w_{1,r} = -1$, and $b_{1,1} = b$ where $b \geq \max_{i,j} \mathbf{U}^{(r)}_{i,j}$; we keep $w_{2,r} = c$ as some large positive constant.

Then $\mathbf{H}^{(1,1)}$ reduces to,

$$\mathbf{H}^{(1,1)} = \text{relu}(-\mathbf{U}^{(r)} + b) = -\mathbf{U}^{(r)} + b \quad \text{since } -\mathbf{U}^{(r)} + b \geq 0 \text{ by definition of b}$$

$$\mathbf{H}^{(2,1)} = \text{relu}(c\mathbf{H}^{(1,1)}_r) \Rightarrow h_{j,k} = c(\max_k(-u_{j,k}+b)) = c(\min_k u_{j,k}+b) \quad \forall u_{j,k} \in \mathbf{U}^{(r)}, h_{j,k} \in \mathbf{H}^{(2,1)}$$

Since $\max_i -x_i + b = \min_i x_i + b$. Thus,

$$\mathbf{f}^{(1)}_i = \text{softmax}(\sum_{k=1}^{n} h_{j,k}) = \text{softmax}\left(nc(\min_k u_{j,k} + b)\right)$$

Therefore, as $c \to \infty$, $\mathbf{f}^{(1)}_i \to f^{maxmin}(i)$ as required.

### 4.9 Max Max Efficiency

**Required:** $f^{\text{max max efficiency}}(i) = \begin{cases} \frac{1}{l} & \text{if } i \in \arg\max_{i \in \{1,\ldots,m\}} \max_{j \in \{1,\ldots,n\}} u^{(c)}_{i,j} + u^{(r)}_{i,j} \\ 0 & \text{otherwise} \end{cases}$

Max Max Efficiency follow from the derivation of Max Max except with $w_{1,r} = 1, w_{1,c} = 1, w_{2,r} = c$ and $b_{1,1} = b$ where $b \geq \min_{i,j}(\mathbf{U}^{(r)}_{i,j} + \mathbf{U}^{(r)}_{i,j})$.

Following the same steps we get,

$$\mathbf{f}^{(1)}_i = \text{softmax}(\sum_{k=1}^{n} h_{j,k}) = \text{softmax}\left(\sum_{k=1}^{n} c(\max_k (u^{(r)} + u^{(c)})_{j,k} + b)\right)$$

$$= \text{softmax}\left(nc(\max_k (u^{(r)} + u^{(c)})_{j,k} + b)\right)$$

$$= f^{\text{max max efficiency}}(i) \qquad \text{as } c \to \infty$$

## 4.10 Minimax Regret

Regret is defined as $r(i,j) = \max_i u_{i,j} - u_{i,j}$ $\qquad u_{i,j} \in \mathbf{U}^{(r)}$

**Required:** $f^{\text{minimax regret}}(i) = \begin{cases} \frac{1}{l} & \text{if } i \in \arg\min_{i \in \{1,\ldots,m\}} \max_{j \in \{1,\ldots,n\}} r(i,j) \\ 0 & \text{otherwise} \end{cases}$

Let $w_{1,rc} = 1$, $w_{1,r} = -1$, and $b_{1,1} = 0$; we keep $w_{2,r} = c$ as some large positive constant.

Then $\mathbf{H}^{(1,1)}$ reduces to,

$$\mathbf{H}^{(1,1)} = \text{relu}(\mathbf{U}_c^{(r)} - \mathbf{U}^{(r)}) = \mathbf{U}_c^{(r)} - \mathbf{U}^{(r)} \quad \text{since } \mathbf{U}_c^{(r)} \geq \mathbf{U}^{(r)} \text{ by definition of } \mathbf{U}_c^{(r)}$$

$$\mathbf{H}^{(2,1)} = \text{relu}(c\mathbf{H}_r^{(1,1)}) \Rightarrow h_{j,k} = c(\max_k(\max_j u_{j,k} - u_{j,k})) \quad \forall u_{j,k} \in \mathbf{U}^{(r)}, \ h_{j,k} \in \mathbf{H}^{(2,1)}$$

Thus,

$$\mathbf{f}_i^{(1)} = \text{softmax}(\sum_{k=1}^{n} h_{j,k}) = \text{softmax}\left( nc(\max_k(\max_j u_{j,k} - u_{j,k})) \right)$$

Therefore, as $c \to \infty$, $\mathbf{f}_i^{(1)} \to f^{\text{minimax regret}}(i)$ as required.

## 4.11 Min Min Unfairness

**Required:** $f^{\text{min min unfairness}}(i) = \begin{cases} \frac{1}{l} & \text{if } i \in \arg\max_{i \in \{1,\ldots,m\}} \min_{j \in \{1,\ldots,n\}} |u_{i,j}^{(r)} - u_{i,j}^{(c)}| \\ 0 & \text{otherwise} \end{cases}$

To represent Min Min Unfairness, we an additional hidden unit in the first layer. Let $\mathbf{H}^{(1,2)}$ be defined in the same manner as $\mathbf{H}^{(1,1)}$.

For $\mathbf{H}^{(1,1)}$, we let $w_{1,r}^1 = 1$, $w_{1,c}^1 = -1$ and $b_{1,1} = 0$ such that,

$$\mathbf{H}^{(1,1)} = \text{relu}(\mathbf{U}^{(r)} - \mathbf{U}^{(c)}) = \max(\mathbf{U}^{(r)} - \mathbf{U}^{(c)}, 0) \quad \text{where the max is applied element-wise}$$

For $\mathbf{H}^{(1,2)}$, we let $w_{1,r}^1 = -1$, $w_{1,c}^1 = -$ and $b_{1,1} = 0$ such that,

$$\mathbf{H}^{(1,2)} = \text{relu}(\mathbf{U}^{(c)} - \mathbf{U}^{(r)}) = \max(\mathbf{U}^{(c)} - \mathbf{U}^{(r)}, 0)$$

Now, notice that if $w_{2,1} = 1$ and $w_{2,2} = 1$,

$$\mathbf{H}^{(2,1)} = \mathbf{H}^{(1,1)} + \mathbf{H}^{(1,2)} = \max(\mathbf{U}^{(r)} - \mathbf{U}^{(c)}, 0) + \max(\mathbf{U}^{(c)} - \mathbf{U}^{(r)}, 0) = |\mathbf{U}^{(r)} - \mathbf{U}^{(c)}|$$

Which gives us a measure of "unfairness" as the absolute difference between the two payoffs.

We can therefore simulate $f^{\text{min min unfairness}}(i)$ by letting $w_{2,1} = -1$ and $w_{2,2} = -1$, and using the output of $\mathbf{H}_r^{(2,1)}$ (which gives us min unfairness), then constructing $\mathbf{f}_i$ by letting $c \to \infty$.

## Footnotes

[1]The claim that we can represent best response by letting parameters tend to infinity may appear dubious given that the models are optimized numerically. However because the $\exp(x)$ function saturates quickly using floating point numbers, in practice $\lambda$ only needs to be moderately large to output a best response.