[Reviews · NeurIPS 2016]

Reviewer 1

Summary

The paper deals with modeling human behavior in game theoretic laboratory settings. Contrary to other related works, this work aims to learn human behavioral models without the assistance of domain experts and (more importantly), without the strong game theoretic assumption of rationality. The paper is of moderate to high relevance as it proposes an improvement to the state of the art where current solutions involve carefully handcrafted representations and solutions. As opposed to this work, where the representation is learned through repeated interactions. The setting and problem is very relevant, as it addresses a problem that will face every AI agent in the real world.

Qualitative Assessment

The technical aspects of the work are sound and explained with a decent amount of detail (much appreciated). The experimental findings lack solid conclusions and need further analysis. Specifically, the experimental setup does not detail what kind of normal form games will be used, the properties of general sum games vary a lot and the implications are profoundly different across classes of games. The results presented do not show much as they are now. With such vague description of the setup 1) results are not reproducible and 2) cannot assess if the proposal generalizes well across classes of games (competitive, cooperative, mixed interest, social dilemmas, coordination/anti-coordination, etc). Furthermore, the analysis of the results is somewhat disappointing. It is very intriguing that the best performing model includes only one action response layer. I wonder if these findings suggest that subjects actually use 1 level models? It would have been informative to comment on how these findings relate to those in behavioral game theory and extend on related results on this field. In conclusion, the work is interesting and I have no question that it advances the state of the art. But the experimental setup, analysis and conclusions could be improved. Typos: - 107. "To apply a such a" - 123. "to represent the all the" - 314. "did not need them to achieve set a new " Detailed questions: - In a footnote you say you ignore salient effects, it would be good if you give some examples of the type of effects. - Section 3. You never explained r is for row player and c for column player. - 226. You say \lambda is a scalar sharpness parameter, why do you need this?

Confidence in this Review

2-Confident (read it all; understood it all reasonably well)


Reviewer 2

Summary

The paper presents a new deep learning architecture for predicting strategic interactions (encoded as normal-form games) between humans.

Qualitative Assessment

The problem is well-motivated and important. The proposed architecture is novel and appropriate for these type of problems and could lead to better prediction of strategic interaction. It is clearly beating the state of the art; though it is not entirely clear how/why. The paper is written well (except for some ambiguities that could be addressed in the response and camera-ready), which helps its potential impact. There are a number of seemingly arbitrary choices made throughout the paper. Both the invariance-preserving hidden units and pooling units seem to remove/restrict strategic reasoning. In some sense, these form a specific class of bounded rationality, but its not intuitive why these operators should be the right ones. Maybe some more motivation (or inspiration from previous models) could help justify these choices. The invariance-preserving restrict the reasoning to always be on the level of the entire matrix. The suggestion that players treat each action similarly is not particularly well-justified; can the authors point to any evidence of this claim in human data? I would expect there to be evidence to the contrary, even in e.g. prisoner's dilemma. The pooling units seem to have the potential to destroy meaningful strategic information. The authors claim to try several (max, mean, sum) in their experiments in the experiments section there is no mention nor comparison there; which one was used? Simplex contraints on parameters? Is this in every layer of the network? Why this specific form of regularization? What happens without it? Could this be the reason that large constants were needed for the softmax? A combination of SGD and RMSProp? Why combine them... and how? What were the values of the learning rate(s)? How long were the models in Figure 4 trained for? Where the larger models trained for longer? It is unfortunate that the action response layers do not contribute much to lowering the loss. Though it is still interesting that 2 AR layers help but then diminish over time. Are these observations consistent with the previous work in this area?

Confidence in this Review

2-Confident (read it all; understood it all reasonably well)


Reviewer 3

Summary

The authors provide a deep learning approach to replicate human behavior in generic normal form games. They provide an interesting review of the preceding literature and show their approach improves upon the state-of-the-art.

Qualitative Assessment

The authors provide a very interesting and generic approach to model human behavior in strategic contexts. it would be interesting to illustrate their approach by analyzing certain applications in more details.

Confidence in this Review

2-Confident (read it all; understood it all reasonably well)


Reviewer 4

Summary

This paper proposed a deep learning method for predicting Human Strategic behavior.

Qualitative Assessment

The model in this paper is very similar to general CNN method and is a common application with CNN.

Confidence in this Review

2-Confident (read it all; understood it all reasonably well)


Reviewer 5

Summary

The authors apply deep-learning to model human behaviour in games (normal-form). They extend the deep-learning approach to exploit the structure of normal-form (invariance-preserving, matrix-units, special pooling-units) and demonstrate an impressive improvement upon prior state of the art.

Qualitative Assessment

Impressive work and it is nice to see deep-learning methodology applied outside the traditional areas. The authors extend several building-blocks to be applicable in their domain and to exploit the underlying nature of normal form games. The results achieved in this paper are an impressive improvement on prior state of the art. Additionally the entire model is learnable and does not require expert knowledge. Because of this the methodology is applicable to a wide range of scenarios as long as they are represented in normal-form. In line 58 I would mention other general deep-learning approaches that depend not on a fixed-dimensional input: e.g. fully convolutional networks. Or if you mean prior models for behavioural modelling they should be cited at this point. For Figure 4 I would like to see the performance of a 100,100,100 model with pooling (or even 50,50,50 & 100,100 with pooling)to further justify the choice of 50,50 and to put the next subfigure into a better context. Lastly the training of the network needs to be discussed in more detail. e.g. choice of parameters and stopping criteria.

Confidence in this Review

2-Confident (read it all; understood it all reasonably well)


Reviewer 6

Summary

The paper suggests a deep learning approach for the task of predicting actions in normal form games. It introduces a new architecture that allows the network to generalize across different input and output dimensions, and show that its performance outperforms that of the previous state of the art, which relies on expert-constructed features.

Qualitative Assessment

In section 3 and in the supplementary material the authors show that by using simple FFNet with permutation they can reach very nice results. The only problem is that FFNets are not invariant to the size of the input payoff matrix. I wonder if they can achieve similar results by padding the input matrix to a constant size that is large enough (by replicating the last column and the last row). Have the authors tried using small Dropout on the input matrix? Have the authors tried 4 layers without using pooling? If using Action Response Layers doesn't help and pooling layers might not be needed then we get a simple CNN with 1x1 convolution kernels. What I am trying to understand is the importance of the proposed new architecture? The text exhibits several typographical errors: line 36: to each player in for each Caption figure 1: Given such a matrix line 123: to represent the all the models

Confidence in this Review

2-Confident (read it all; understood it all reasonably well)